# Quality of *Tenebrio molitor* Powders: Effects of Four Processes on Microbiological Quality and Physicochemical Factors

**DOI:** 10.3390/foods12030572

**Published:** 2023-01-28

**Authors:** Xin Yan, Sophie Laurent, Isabelle Hue, Sylvie Cabon, Joelle Grua-Priol, Vanessa Jury, Michel Federighi, Geraldine Boué

**Affiliations:** 1Oniris, INRAE, SECALIM, 44300 Nantes, France; 2Oniris, Université de Nantes, CNRS, GEPEA UMR 6144, 44322 Nantes, France; 3MicroBioTech, Oniris, 44322 Nantes, France; 4EnvA, ANSES, LSA, 94700 Maison-Alfort, France

**Keywords:** entomophagy, edible insect powder, microbiological risks, gross composition, predictive microbiology, HACCP strategy, heat treatments

## Abstract

*Tenebrio molitor*, the first edible insect approved as a novel food in the EU, is a promising candidate for alternative protein sources, implementing circular and sustainable production systems. This study aims to determine the microbiological quality and physicochemical properties of mealworm powders obtained by four different processing pathways. Contents of dry matter, protein, fat, ash, water activity (aw) and a range of microbial counts were measured and analyzed by one-way ANOVA with Tukey’s test. Results showed small differences in the proximate composition of the powder samples (protein 55.62–57.90% and fat 23.63–28.21% of dry matter, DM), except for the one that underwent a defatting step (protein 70.04% and fat 16.84%), *p* < 0.05. A level of water activity of less than 0.2 was reached for all pathways. Fresh mealworm samples had high total aerobic counts (8.4 log CFU/g) but were free of foodborne pathogens. Heat treatments applied during transformation were sufficient to kill vegetative cells (reduction of 2.8–5.1 log CFU/g) rather than bacterial endospores (reduction of 0.3–1.8 log CFU/g). Results were confirmed by predictive microbiology. This study validated the efficacy of a boiling step as critical control points (CCPs) of insect powder processing, providing primary data for the implementation of HACCP plans.

## 1. Introduction

The practice of eating edible insects (entomophagy) has been part of the regular diet for millions of people in various regions in Asia, Africa and Latin America for a long time [1]. In Europe, insect consumption as food is relatively new but has recently sparked growing interest owing to an urgent switch to sustainable food production and protein source. Compared to conventional animal husbandry, rearing insects has a smaller ecological influence due to their lower emission of greenhouse gases and ammonia per kg mass gain [2]. Moreover, the feed conversion ratio is higher, and the production duration is much shorter than the breeding of other animals [3]. From a nutritional perspective, edible insects are recognized as an excellent source of essential nutrients, including protein, essential fatty acids, vitamins and some mineral elements. Huge variations exist in the nutritional composition of insects and depend on the species, developmental stages, habitat and feeds. For example, the protein content of edible insects varies from 20 to 76% of dry matter, which is similar to that of poultry, pork, beef and fish and even higher than that in soybean; the fat content varies from 2 to 50% of dry matter and presents a saturated/non-saturated fatty acid ratio below 40%, which is better than that observed in fish or chicken [4,5]. Moreover, edible insects can provide superior health benefits due to their high levels of essential amino acids (EAA), omega-3 and omega-6 fatty acids, vitamin B-12, iron, zinc, fiber and antioxidants [6,7]. For these reasons, the last decade has witnessed an exponential interest in insects as food, as evidenced by the number of publications and the number of estimated insect producers worldwide [8]. 

However, the insect sector is still in its early stage, especially in European countries. The report of the Food and Agriculture Organization (FAO) stated that some challenges of the insect sector, such as potential food safety issues, the establishment of suitable regulatory frameworks, research gaps, upscaling the production and negative attitudes of consumers, need to be overcome to have a more global reach [9]. For example, according to the EU’s Novel Foods Regulation (EU N°2015/2283) and general food law, authorization to place insects on the market must be based on a risk assessment regarding all the potential hazards. In 2015, the European Food Safety Authority (EFSA) promoted a scientific opinion on the risk profile related to the production and consumption of insects as food and feed [10]. The opinion explored the potential safety issues (biological, chemical and environmental) associated with the entire insect production chain. More precisely, microbial hazards would be one of the major safety concerns as insects are rich in nutrients and moisture, providing favorable conditions for microbial survival and growth [11]. 

Recent studies have shown that edible insects generally contain complex ecosystems with marked variations in microbial load and diversity as well as stable and species-specific microbiota [12]. High levels of mesophilic aerobes, bacterial endospores or spore-forming bacteria, Enterobacteriaceae, lactic acid bacteria, psychrotrophic aerobes, yeasts and molds have been found in raw insects [13]. Among these, potentially harmful microbes (i.e., pathogenic, mycotoxigenic and spoilage microbes) may also be present. Kooh et al. [14] identified several significant microbial hazards in insect powders based on a two-dimensional Risk Matrix (multiplication of Severity index and Likelihood index), including *Salmonella* spp., STEC, *L. monocytogenes*, *Cronobacter* spp., *C. perfringens*, *C. botulinum*, *S. aureus*, *B. cereus* group. However, currently, the amount of the literature regarding the microbial quality and potential hazards of different insect species is still rather limited; more quantitative research should be carried out in order to implement an appropriate HACCP (Hazard Analysis Critical Control Point) system and progress in its risk assessment for edible insect supply chains. 

*Tenebrio molitor* larva (yellow mealworm, hereinafter referred to as mealworm) is viewed as one of the most promising species due to its nutritional value and ability to mass-produce [15]; moreover, it is one of the most often raised species in Europe. In the current European market, the best-selling products are whole insects and insect meal (in the case of protein flour or powders obtained by grinding insects), mainly based on *T. molitor* or *Acheta. domesticus* [16]. In 2020, the EFSA Panel on Nutrition, Novel Foods and Food Allergens (NDA) delivered an opinion on the safety of dried yellow mealworm as a novel food (NF) pursuant to Regulation (EU) 2015/2283 [17]. Soon after, the authorization for *T. molitor*, the first insect species as a novel food in the EU was issued in June 2021 [18]. Consequently, mealworms in the form of “flour or powder” is perhaps an appealing choice to improve customer acceptance and maximize market opportunities. 

Well-designed manufacturing processes are required to produce high-quality insect products while maintaining food safety to meet the growing interest of consumers. Various technologies have been employed to process insects into non-recognizable forms, such as powders or flour, including heat treatment (e.g., boiling, blanching, cooking), drying (e.g., sun-drying, freeze-drying, oven-drying), grinding (wet or dry) and some innovative methods mainly for protein, fat and/or chitin extraction [19]. With regard to microbial inactivation, traditional thermal treatments are still the predominant methods in the insect sector [20]. While freezing is considered an ideal method to kill insects, it is the similar way of killing in nature as well as good instant preservation and storage methods [21]. According to the EFSA and the International Platform of Insects for Food and Feed (IPIFF), freeze-drying represents the current industrial practice for European producers of drying insects [10,22]. This technique uses a vacuum to remove moisture from a frozen product by sublimation and usually maintains a very good quality and causes the least damage to proteins [23]. Several studies explored the effect of heating treatments on the microbial load and pathogenic microorganisms of insects [24,25] and the influence of different drying methods on their nutrient quality [26,27]. However, considering the different processing pathways (combination of different steps) to produce insect powder, little is known or available about their impacts on microbial quality and physicochemical properties. 

Hence, the present study selected four different mealworm powder processing pathways, including boiling or freezing slaughter, oven- or freeze-drying, defatting or not and grinding. These pathways, called A, B, C and D, are described in Figure 1. The selection of these treatments was based on current industry practices and ongoing research interests identified by Yan et al. [20]. The aim is to investigate their impacts on microbial quality and physicochemical properties of mealworms. Contents of dry matter, protein, fat, ash and water activity (a_w_) were measured, and the results were analyzed by one-way analysis of variance with Tukey’s test. In addition, a microbiological predictive model was used to assess the efficacy of heat treatment steps on eight significant hazards. The results of this study can provide preliminary data and proof for the application of the HACCP system in the insect food industry.

## 2. Materials and Methods 

### 2.1. Raw Materials 

Live yellow mealworm larvae (two batches), which already had a 24 h period of fasting, were purchased from a French company (France Insectes^®^, Le Gallet, France) that specialized in rearing insects for pet food. They were kept in plastic boxes and transported in hygienic conditions to the laboratory. Then, they were divided randomly into two groups: one group was immediately stored in the refrigerator at 4 °C before the first processing trial, and the other group was frozen for one week to duplicate the experimentation. 

### 2.2. Insect Powder Processing Trial

The four processing pathways (PA, PB, PC, PD) are described in Figure 1 and Table 1. During the trials, mealworm larvae samples were taken aseptically and ground with a blender (Russell Hobbs, Ellwangen, Germany), resulting in homogenous mixtures for further analysis. Two ways of slaughter were employed, either by boiling or freezing (only in pathway D). The boiling and cooking step (pathway B) were performed in a thermostatically controlled double-walled tank Thermomix^®^ (Vorwerk, Nantes, France) with a temperature sensor. The mealworm paste, after cooking, was transferred to a centrifuge (Beckman Coulter, Indianapolis, IN, USA) for centrifugal separation to remove oil layer. 

Oven-drying steps were conducted in pathways A and B and stopped only when water activity was below 0.5 (Samples PA-OD and PB-OD were taken to measure a_w_ each hour). While freeze-drying was applied to whole mealworms of pathways C and D in freeze-drying equipment (FTS Systems, New York, NY, USA). Finally, all the pathways contained an identical grinding step by a blender (Russell Hobbs, Ellwangen, Germany) to obtain different mealworm powders. 

### 2.3. Proximate Composition Analysis and Water Activity Determination

#### 2.3.1. Proximate Composition Analysis

The dry matter content was determined gravimetrically by drying at 105 °C to a constant weight [28]. Protein and ash content were determined according to standard AOAC (Association of Official Analytical Chemists) methods (i.e., AOAC 968.06 and AOAC 923.03, respectively [29]. The protein content was determined by the Dumas method using an FP828P carbon/nitrogen analyzer (LECO, St. Joseph, MI, USA) using the conventional nitrogen-to-protein conversion factor of 6.25. It should be noted that this factor can overestimate the protein content due to the presence of non-protein nitrogen in insects such as chitin [30]. The ash content was determined by incineration in a muffle furnace (Nabertherm, Lilienthal, Germany) at 550 °C for 6 h. The crude fat content was determined using an accelerated solvent extraction unit, Thermo Scientific™ Dionex™ ASE™ 150 (Thermo Fisher Scientific, Courtaboeuf, France), according to the method developed by Wei et al. [31] with slight modifications. The extraction cell (35 cm^3^) in stainless steel was fitted with a cellulose filter and a stainless-steel frit, both at the top and the bottom, and was filled with a sample (5 g) and diatomaceous earth (Thermo Scientific™ Dionex™ ASE™). The extraction process consisted of the following steps: loading the cell into the oven (maintained at 120 °C), filling the cell with solvent (n-hexane), heating the cell (7 min/extracting cycle, 10 MPa +/− 2 MPa) and extracting twice. Nitrogen gas was employed to collect the remaining extract into the collection bottle (250 cm^3^) at the end of the extraction before unloading the cell. The oil/solvent mixture was evaporated under vacuum to dryness at 40 °C in a rotary evaporator. Carbohydrates were calculated by the difference between the total mass and the sum of crude protein, lipid content, moisture and ash.

#### 2.3.2. Water Activity Determination

Water activity (a_w_) is a measure of the availability of water to take part in chemical reactions (varies from 0–1.0), determining the limits for microbial growth [32]. For example, most bacteria need a_w_ > 0.91, and most molds need a_w_ > 0.80; and once a_w_ < 0.6, the product is generally considered to be shelf-stable [33]. Water activity (a_w_) was measured on a 2–3 g aliquot of each sample using an AquaLab Pre a_w_ meter (Decagon Devices Inc., Pullman, DC, USA) until a_w_ and temperature were stable during 5–15 min. Larvae samples were ground in a blender before being analyzed. 

#### 2.3.3. Statistical Analysis

The values obtained from three repetitions were expressed in means ± standard deviation. The data were subjected to one-way analysis of variance (ANOVA) and Tukey’s test procedures with a level of significance set to 0.05. All the statistical analyses were performed with XLSTAT software version 2018.5 (Addinsoft).

### 2.4. Microbiological Analysis

A number of classic microbiological indicators, such as total aerobic count (TAC), *Enterobacteriaceae*, lactic acid bacteria (LAB), bacterial endospores, yeasts and molds count (YMC), are widely used as measures of hygienic conditions or quality of food products. The fresh mealworm samples were taken aseptically (60 g for each) and transported to the laboratory at ambient temperature and dealt with upon arrival (<10 min). All the samples from power processing trial were stored in refrigerator at 4 °C (<24 h) before analysis. The microbial analysis methods are displayed in Table 2. All microbial counts were expressed as log CFU/g. 

Quantification of TAC, LAB and bacterial endospores were performed in our laboratory following next steps, while others were outsourced to an accredited laboratory. Additionally, 10 g of subsamples were taken aseptically from each step during powder processing and mixed with 90 g of peptone water (PPS, 0.85% NaCl, 0.1% peptone, Biokar Diagnostics, Beauvais, France) in a stomacher plastic bag. After homogenization for 60 s in a paddle blender (Bagmixer^®^ Interscience, Saint Nom, France), the primary suspensions were stored at ambient temperature for at least 1 h. Ten-fold dilution series (from 10^−1^ to 10^−6^) were prepared and plated on different media using the pour-plate technique, according to the ISO standards. TAC were assessed after incubation on Plate Count Agar (PCA, Biokar diagnostics, Solabia Group, PANTIN, France) at 30 °C for 72 h; bacterial endospores were determined on PCA plus starch at 30 °C for 72 h after a heat shock treatment (water bath at 80 °C for 10 min) of the ten-fold dilutions; LAB were incubated on de Man, Rogosa Sharpe agar (MRS, Biokar diagnostics, Solabia Group, PANTIN, France) at 30 °C for 72 h. For each dilution, two replicates were performed. All microbial counts were expressed as log CFU/g. 

### 2.5. Predictive Microbiology Modeling

In order to assess the efficacy of heat treatments during powder processing, this study estimated log reductions of eight significant hazards based on Bigelow model. D-value is the time required at a given temperature or set of conditions to achieve a log reduction [34], and z-value is the increase of temperature necessary to achieve a tenfold reduction in D-values [35]. Whereas D-value reflects resistance of a microorganism to a specific temperature, z-value provides information on the relative resistance of a microorganism to different destructive temperatures, allowing for the calculation of equivalent thermal processes [36]. Once D-value at a certain reference temperature (D_ref_) is known, D-values can be estimated for every desired temperature (see Equation (1)) [37]: logD = logD_ref_ − (T − T_ref_)/z,(1)
where D is the time to achieve a log reduction at temperature T (in minutes); D_ref_ is the D-value at T_ref_ (in minutes); T_ref_ is the reference temperature (in °C); z is the temperature increase (in °C) needed to reduce the D-value with a factor of 10.

Reference D-value and z-value data of eight hazards were collected from the scientific literature (Table 3); they were related to certain temperatures and various products or tested media. Then the D-values at 80 and 100 °C were calculated, respectively. Based on these D-values, the log reductions achieved by two heat treatments (5 min at 100 °C and 30 min at 80 °C) can be estimated.

## 3. Results

### 3.1. Microbiological Status of Fresh Mealworms (Initial Level)

Average microbial counts of fresh mealworm larvae were relatively high (see Appendix A), with mean TAC 8.4 log CFU/g, 8.5 and 8.3 log CFU/g for batches 1 and 2, respectively. LAB counts ranged from 7.8 to 8.1 log CFU/g, >4.2 log CFU/g for *Enterobacteriaceae* and 4.0 log CFU/g for yeasts and molds. Lower numbers were observed for bacterial endospores 3.4−5.3 log CFU/g. Similar microbial counts were found in rinsed fresh mealworm samples.

Overall, *L. monocytogenes, Salmonella* spp. and *C. sakazakii* were not detected from any of the analyzed samples (see Appendix A). Almost all the other hazards were below the limit of quantification considering the dilution used (<1.0 log CFU/g; <2.0 log CFU/g for coagulase-positive *Staphylococci*), except for sulfite-reducing anaerobes at low-level (1.0 log CFU/g), which is an indicator for *Clostridium* spp., mainly *C. perfringens.*

### 3.2. Microbiological Status of Mealworm Powders (End Products)

For mealworm powders obtained from processing trials, the pathways involved in at least one heat treatment (PA, PB, PC) showed a better microbial quality (e.g., TAC 4.4−5.7 log CFU/g) than raw mealworms. Likewise, lower counts were found in bacterial endospores (2.1−2.7 log CFU/g), LAB (1.8−5.2 log CFU/g), *Enterobacteriacae* (<1.0−3.6 log CFU/g), yeasts and molds (<1.0−3.4 log CFU/g) compared to fresh mealworms. However, the powder from pathway D (PD-P) was highly contaminated, similar to fresh mealworms (e.g., TAC 8.5−8.9 log CFU/g, bacterial endospores 4.4−5.2 log CFU/g, LAB 7.7−8.3 log CFU/g, *Enterobacteriacae* > 4.2 log CFU/g, yeasts and molds 2.6−2.9 log CFU/g) (see Appendix A). Additionally, all the significant hazards were below the limit of quantification or not detected in the analyzed powder samples (see Appendix A).

### 3.3. Microbial Inactivation Effects of Processing Steps

As Figure 2a–c indicates, after boiling (100 °C for 5 min), LAB counts reduction was evident (4.9−5.0 log CFU/g), followed by TAC (2.8−3.1 log CFU/g), whereas cooking (80 °C for 30 min) also had effects on LAB (>1.9 log CFU/g) and TAC (3.1 log CFU/g) (Figure 2b). On the other hand, the reductions in bacterial endospore counts were relatively lower, 1.3−1.8 log CFU/g for boiling and 0.3 log CFU/g for cooking. A slight increase in microbial counts (0.1−2.5 log CFU/g) after heat treatments was also worth noting, partly due to the drying steps that concentrate bacteria per gram in powders. In Figure 2d, microbial counts (TAC, LAB, bacterial spores) during the processing trial had no significant change after freezing slaughter (−18 °C) and freeze-drying. All the results above were mean values of two batches.

Figure 3 shows the estimated log reduction of significant hazards after different heat treatments using predictive microbiology models. The heating applied in processing trials led to a 0.01–6.28 log reduction for spore-forming bacteria and *S. aureus* enterotoxin, which have a higher heat resistance (longer D-values) than vegetative form bacteria, while the latter all showed an enormous log reduction (>346.52, see Appendix A). In general, boiling at 100 °C for 5 min has a better effect on microbial inactivation than cooking at 80 °C for 30 min.

### 3.4. Physicochemical Properties 

The water activities and dry matter of fresh and powdered mealworms are shown in Table 4. The a_w_ of fresh mealworm is relatively high (0.99) (moisture content corresponding to 70.5%) as well-suitable for microbial growth and enzymatic and chemical activity [47]. Each processing pathway was comprised of a drying step (oven or freeze-drying), and the a_w_ of all samples were reduced until at very low levels (0.08 for powder A, 0.16 for powder B and 0.17 for C and D, corresponding to a moisture content between 2.05% and 4.64% wet basis).

The proximate compositions of fresh and powdered mealworms are shown in Table 4. Fresh mealworm larvae contained 29.5 g/100 g of dry matter (DM), including 66.1% of crude protein (19.5% of total), 19.9% of crude fat (5.9% of total) and 4.6% of ash (1.4% of total). Mealworm powders without oil extraction (A, C, D) contain significant amounts of protein (55.6–58.4% of DM), which even increases to 70.0% in defatted powder B. The fat contents of powder A, C and D are between 23.6–28.2% based on a DM basis, and as expected, the fat content of the defatted powder (B) was significantly lower (16.84% of DM). The dry matter of powdered mealworm varied significantly with the processing pathway (97.95%, 97.87%, 95.54% and 96.83% for powder A, B, C and D, respectively). 

## 4. Discussion

### 4.1. Microbiological Status of Mealworms

Fresh or unprocessed mealworm larvae contain relatively high levels of microbial loads, which was in line with those reported in previous studies, including TAC 6.4–9.3 log CFU/g, LAB 4.9–8.3 log CFU/g, *Enterobacteriaceae* 5.0–7.7 log CFU/g, bacterial endospores or spore-forming bacteria 0–5.3 log CFU/g, psychrotrophic aerobic count 5.9–7.6 log CFU/g and yeasts and molds 2.6–6.5 log CFU/g [20]. Mealworms, after rinsing, did not show many differences in microbial counts. This can be explained by the fact that insect microorganisms mainly originate from intestinal tracts, and insects are usually not degutted. Possible contaminations occur during the rearing environment and conditions (e.g., substrate and feed), unhygienic handling and storage; for instance, mealworm larvae stay in close contact with their feces before sieving. In addition, two investigated batches showed similar microbial counts, as they originate from the same company with similar rearing cycles. 

The absence of microbial pathogens in analyzed fresh mealworm samples may be due to the modern rearing techniques and GHPs of the company. To date, common foodborne pathogens (e.g., *L. monocytogens*, *Salmonella* spp., *E. coli*) were not detected in other culture-dependent analyses. However, in some studies [48,49], *Staphylococcus* spp. was found in a small number of samples (3.7–6.0 log CFU/g) in fresh mealworms. Garofalo et al. [50] demonstrated that mealworms contained a low abundance (below 2%) of *Listeria* spp., which was not possible to identify at the species level. *L. monocytogens* [51], and *Salmonella* spp. [52] also can survive and grow in the rearing substrate of mealworms.

Concerning mealworm powder, very little data on pathogenic microorganism detection are available, but spore-forming bacteria such as *Bacillus* spp. and *Clostridium* spp. have been detected in processed mealworms (boiled and dried) [53,54]. The likelihood of the occurrence of *B. cereus* spores in dried and powdered insects is moderate, occurring regularly [55].

In Europe, no specific microbiological criteria for insect-based products are established for human consumption currently. Some food safety authorizes—such as the Belgian Superior Health Council (BSHC), the Federal Agency for the Safety of the Food Chain (FASFC) and the Netherlands Food and Consumer Product Safety Authority (NVWA)—have composed an opinion to refer to the hygiene criteria for minced meat in EU Regulation (EC) No. 1441/2007 for edible insects [56,57]. For example, The TAC limit criteria for minced meat is 5.7–6.7 log CFU/g, with at most two of five sample units tested from food batches giving values between the range [58]. In the present study, the TAC of fresh mealworm (8.4 log CFU/g) significantly exceeds the criteria for minced meat. 

### 4.2. Evaluation of Identified Critical Control Points (CCPs) 

Heat treatments such as boiling, blanching, frying, roasting and other cooking methods are the most commonly applied steps for microbial inactivation (CCPs). Their efficacy in improving microbial quality depends on the combination of duration and temperature as well as the characteristics of microorganisms. There is no doubt that boiling (100 °C for 5 min) applied in pathways A, B and C and cooking (80 °C for 30 min) of pathway B are the major CCPs in our processing trials. As confirmed by the present study, similar microbial reductions after heat treatments were observed in other studies [20]: TAC (3.5–5.0 log CFU/g), *Enterobacteriaceae* (4.0–5.0 log CFU/g), LAB (4.0–5.5 log CFU/g), psychrotrophic aerobic count (4.5–6.0 log CFU/g) and yeasts and molds (2.0–4.0 log CFU/g). Aside from some exceptions, these indicators are relatively sensitive to heat treatments. Klunder et al. [11] concluded that a heating step (i.e., boiling for 10 min) is sufficient for the inactivation of *Enterobacteriaceae* in insects, but some bacterial endospores will survive. Vandeweyer et al. [59] explored the effect of blanching and microwave drying on microbial loads of mealworm larvae. Regardless of treatment times (10 s, 20 s, 40 s), blanching alone or blanching plus microwave drying represent a heat treatment that kills vegetative cells but no or hardly any spores. Caparros Megido et al. [25] demonstrated that household cooking techniques such as vacuum-cooking (74 °C for 1 h), pan-frying (1 min) and boiling (100 °C for 1 min) decreased the TAC level to below the lower limit for minced meat (5.7 log CFU/g). 

Furthermore, the results of predictive models provided a general trend of thermal inactivation performance on significant pathogens since the parameters used are not specific to insect matrices. Particular attention should be paid to spore-forming bacteria (e.g., *Bacillus* spp., *Clostridium* spp.) as well as different heat-resistance of strains because they can survive mild thermal treatments (<100 °C) and exist in the final products. Garofalo et al. [50] reported that *C. perfringens* spores had been found in marketed whole processed (boiled and dried) crickets, grasshoppers and mealworms. Caparros Megido et al. [60] concluded that sterilization was the most efficient treatment for reducing microbial counts in mealworms, which is known to kill bacterial spores. In the food industry, to achieve sterilization, a higher temperature (usually 121–134 °C) is required by saturated steam or autoclave, which imposes some negative effects on the foodstuff, such as organoleptic properties loss and nutritional substances damage [61]. Therefore, the effective and acceptable temperature–time combinations of heat treatment regarding the significant hazards of insects need to be further investigated.

Drying is primarily a preservation step to extend shelf life. However, it also makes food products more efficient and cost-effective to store (they do not need to be refrigerated or kept cool) and ship (powders weigh less) [21]. The oven-drying step applied in pathways A and B, which supply mild heat (<100 °C), may bring additional microbial reduction, but it remains uncertain given the decrease of a_w_ during drying also increases the resistance of microorganisms [62]. In addition, low a_w_ of dried insects contributes to inhibiting the presence or growth of microorganisms, so the inactivation effects of the hot drying step (as a CCP) in this processing trial still need further research.

### 4.3. Evaluation of Four Processing Pathways 

Processing pathways A, B and C provided a similar inactivation performance on the final powders due to the adopted heat treatments, particularly pathway B, which contained two heating steps. For instance, the TAC of the powders A, B and C is 4.7–5.5 log CFU/g, all below the lower limit (5.7 log CFU/g) for minced meat. The prerequisite for achieving desired effects is to carefully control the temperature and duration to meet the critical limits (e.g., the core temperature reaches 100 °C). However, these treatments are insufficient to eliminate bacterial spores and *S. aureus* enterotoxin when they exist in insects. Of note, the slight increase in microbial counts (Figure 2) after heat treatments may be caused by sample concentration during drying or recontamination afterward. 

As expected, pathway D, which only contains freezing and freeze-drying, is considered inappropriate for producing microbial-safe insect powders. Bußler et al. [63] also reported that freeze-drying and grinding of the *T. molitor* larvae at low temperatures led to an insignificant log-reduction of the TAC by 0.1(±0.1). However, freezing and frozen storage have a variable effect on microorganisms; for instance, quick freezing and thawing would result in higher microbial survival rates than slow ones; relatively higher storage temperatures (−4–−10 °C) have a greater lethal effect on microorganisms than lower temperatures (−15–−30°C) [64]. Freeze-drying is known to mostly inhibit microorganisms’ growth rather than kill them, and when rehydrated, many could return to a vegetative form that might be harmful [55]. So, in order to provide safe products, the initial quality of raw materials is crucial; good agricultural practices (GAPs) must be implemented during primary production, avoiding contamination from the farming environment (e.g., soil, dust, manure, substrate), feed and water used for rearing, unhygienic handling of workers and so on. It is also imperative to process insects in a controlled, traceable, strict containment system that employs sanitary techniques, Good Hygiene Practices (GHPs) and Good Manufacturing Practices (GMPs), which will help to reduce the occurrence of hazards. Otherwise, it is recommended to combine pathway D with other microbial inactivation methods to control significant hazards.

### 4.4. Physicochemical Properties of Fresh and Powdered Mealworms 

All processing methods applied, as they include a drying step, significantly decreased the water activity a_w_, from an initial value of 0.99 for fresh mealworms to very low levels (a_w_ < 0.2), regardless of the pathway considered. This a_w_ level is sufficient to inhibit microbial growth; however, at very low water activity (a_w_  <  0.2), lipid oxidation that is responsible for rancidity and the development of off-flavors can increase [65].

Regarding the proximate composition, the fat, ash and carbohydrates contents of fresh mealworms were similar to values reported in other studies, whereas the protein content (66% of the DM; kp = 6.25) was higher than the values commonly found (usually between 46 and 56% of the DM; kp = 6.25) [15,53,66,67,68]. However, the protein content of mealworms and, more widely, of insects can considerably vary depending on their diet [67,68]. Mealworm larvae fed on milk-based feed can, for example, reach a protein content of 68% on a dry matter basis [68]. According to the results of ANOVA, all the processing pathways have reduced the protein content compared to fresh mealworms, except for pathway B, where an increase in protein content was observed and can be easily explained by the defatting step. The decrease in protein content in the case of pathways A (boiled and oven-dried) and C (boiled and freeze-dried) can probably be attributed to the boiling step, where the dissolution of soluble proteins in boiling water can occur. These results are in accordance with previous findings in edible insects [69,70,71]. A significant loss of protein was also observed when pathway D was applied (freezing and freeze-drying), while several studies showed no impact of freezing as a killing step or freezing-dying on the proximate composition of insects [27,72,73,74]. Differences in insect preparation, handling and protein determination are the likely causes of these diverse outcomes. Any of the other parameters, such as fat, ash and carbohydrates content, were similar, despite small fluctuation ranges.

In terms of overall results, processing pathway D was the less efficient method for its inability to preserve protein in mealworms compared with the others processes. Although the protein content of mealworm powders seemed to be similar between processing pathways A, C and D, their effects on protein quality remain unclear, such as denaturation, modification or destruction of amino acids and the Maillard reaction. 

## 5. Conclusions

In the present study, knowledge of microbiological quality and the main physicochemical factors of fresh and powdered mealworms was obtained thanks to the combination of processing trials, analysis and predictive microbiology. The high level of microbial contamination of fresh mealworms highlights the necessity of a processing pathway involving an effective microbial inactivation, such as a boiling step considered as a CCP. Specific attention must be paid to bacterial spores when establishing the microbial criteria in insect-based products, such as insect powder. Additionally, the nutritional value of all the powdered insects remained high despite the loss of proteins in processing pathways A, C and D. Nevertheless results of this study must be considered in light of the limited number of samples analyzed. Therefore, further studies are needed to collect more data and go further with the implementation of culture-independent analysis and non-thermal inactivation methods. More insights concerning the effects of different processing steps on protein quality, such as denaturation, modification or destruction of amino acids and digestibility, are also necessary. On a broader scale, the next step in the development of the edible insect sector seems to be the enhancement of consumer acceptance by developing safe and high-quality insect food products.

## Figures and Tables

**Figure 1 foods-12-00572-f001:**
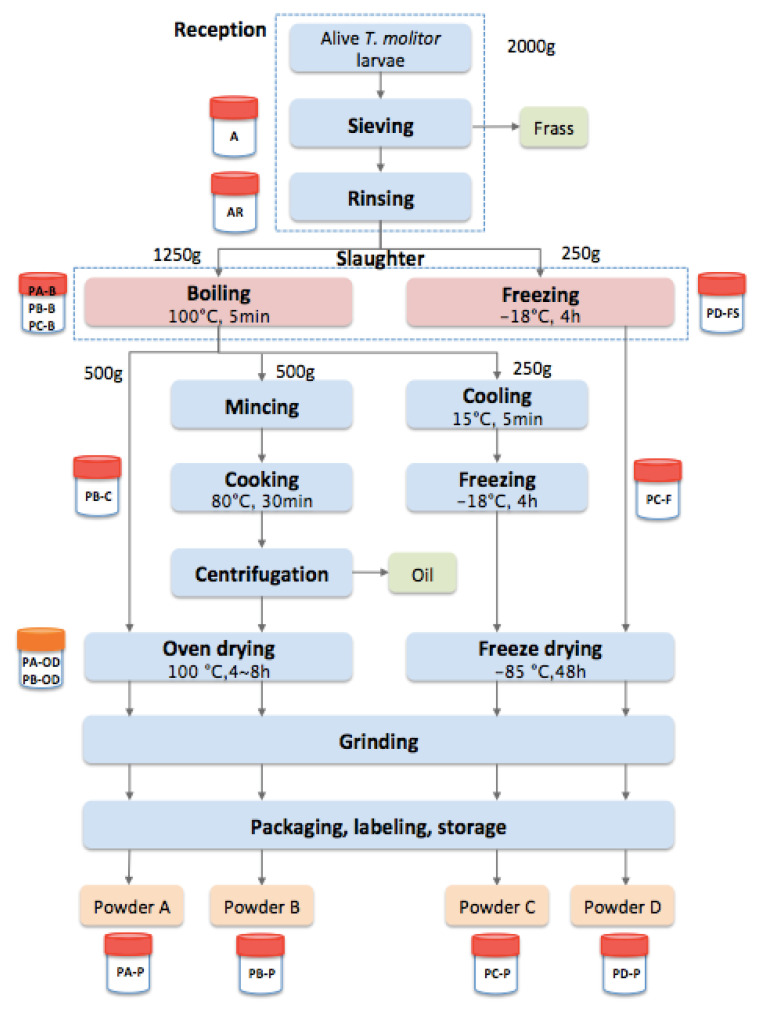
Flowchart of mealworm powder processing pathways and samples collected (adapted from [14]).

**Figure 2 foods-12-00572-f002:**
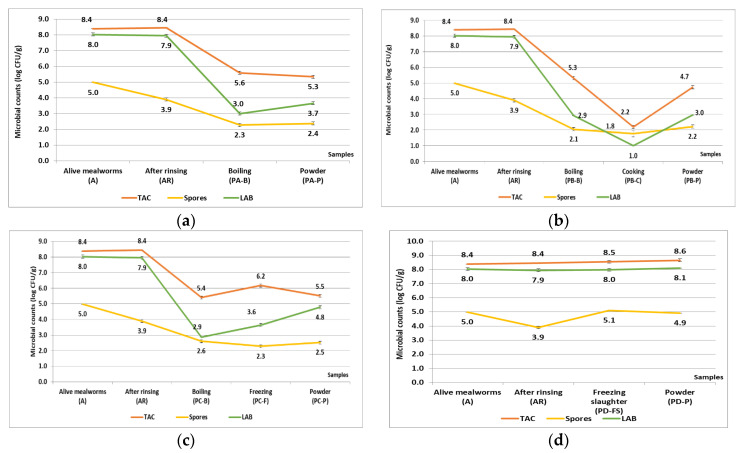
Microbial counts variation of processing pathway A (**a**), pathway B (**b**), pathway C (**c**), pathway D (**d**) (TAC: total aerobic viable count; spores: bacterial endospores; LAB: lactic acid bacteria; data are mean values of two batches).

**Figure 3 foods-12-00572-f003:**
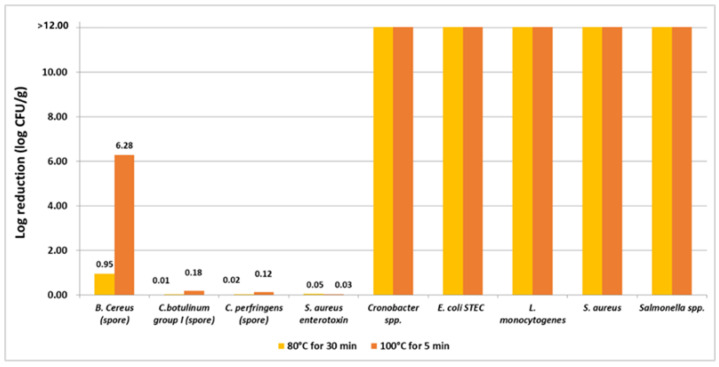
Log reduction of significant hazards estimated with D_ref_ values for heat treatments.

**Table 1 foods-12-00572-t001:** Description of four processing pathways and samples collected (adapted from [14]).

Processing Step	Description	Processing Pathway				Sample
		A	B	C	D	
**Sieving**	Sieving was performed to remove residues of substrates, frass and dead larvae.	+	+	+	+	Alive (A)
**Rinsing**	Mealworm larvae was carefully rinsed with clean water.	+	+	+	+	After rinsing (AR)
**Boiling**	Mealworm larvae were boiled in hot water (insect water ratio 1:1) for 5 min (timer started only when water temperature reached 100 °C after the addition of larvae) and then drained.	+	+	+		PA-B PB-B PC-B
**Freezing slaughter**	Larvae were put in a thin layer of less than 5 cm into freezer for 4 h at −18 °C.				+	PD-FS
**Mincing**	Mincing was performed at speed 10 with a multifunctional appliance Thermomix^®^.		+			
**Cooking**	Cooking was performed in Thermomix^®^ tank at 80 °C for 30 min (insect water ratio 1:1).		+			PB-C
**Centrifugation**	Centrifugation was applied to separate the oil and paste (5 min 80 °C with a force of 5000 g).		+			
**Cooling**	Boiled mealworms were put in a cold-water cooling system at 15 °C for 5 min.			+		
**Freezing**	Cooled mealworms were frozen for 4 h at −18 °C.			+		PC-F
**Oven-drying**	Whole insects or insect paste were dried in a fine layer of a laboratory oven at 100 °C for 4–8 h.	+	+			PA-OD PB-OD
**Freeze-drying**	Frozen mealworms were put into freeze-drying equipment at −85 °C for 48 h.			+	+	
**Grinding**	Insect powder was obtained by grinding the obtained larvae with a blender.	+	+	+	+	
**Packaging,** **Labeling,** **Storage**	All the powders were packaged into plastic bottles, labeled and stored at ambient temperature.	+	+	+	+	PA-P, PB-P, PC-P, PD-P

+ means that the step was applied in the processing pathway A, B, C or D.

**Table 2 foods-12-00572-t002:** Microbiological analysis methods.

Indicators/Targets	Methods
Bacterial endospores	Plate Count Agar (PCA) with 0.2% starch, incubated at 30 °C for 72 h after thermal treatment at 80 °C for 10 min
*Clostridium perfringens*/g	NF EN ISO 7937
Coagulase-positive staphylococci (37 °C)/g	Internal method adapted from NF V 08-057-1
*Cronobacter* spp./10 g	NF EN ISO 22964
*Enterobacteriaceae* (30 °C)/g	NF * V 08-054
*Escherichia coli* β-glucuronidase positive/g	Internal method adapted from NF ISO 16649-2
Lactic acid bacteria (LAB)	De Man, Rogosa Sharpe (MRS) agar, incubation at 30°C for 72 h
*Listeria monocytogenes*/25 g	AES 10/03-09/00 ***
Presumptive *Bacillus cereus* (30 °C)/g	BKR 23/06-02/10 **
*Salmonella* spp./10 g	BRD 07/11-12/05 ****
Sulfite-reducing anaerobes (*Clostridium* spp.) (46 °C)/g	Internal method adapted from NF V 08-061 box
Total aerobic count (TAC)	Plate Count Agar (PCA), incubation at 30 °C for 72 h
Yeasts and molds (on products at a_w_ < 0.96)/g	NF V 08-036

* NF: Norme Française; ** refer to COMPASS^®^ *Bacillus cereus* Agar: alternative analysis method validated by AFNOR (Association Française de Normalisation) Certification for the enumeration of *Bacillus cereus*; *** refer to ALOA^®^ ONE DAY: alternative analysis method validated by AFNOR Certification for the detection of *Listeria* spp. and *Listeria monocytogenes*; **** refer to RAPID’Salmonella: alternative analysis method validated by AFNOR Certification for the detection of *Salmonella* spp.

**Table 3 foods-12-00572-t003:** Inactivation parameters of eight selected significant biological hazards (a_w_ > 0.9).

Biological Hazards	Thermal Inactivation Parameters			References
	T_Ref_ (°C)	D_Ref_ * (min)	z (°C) *	
*B. Cereus* (spore)	95	2	(8−12.5)	[38]
*C. botulinum* Type I (spore)	121.1	0.21	10	[39]
*C. perfringens* (spore)	100	(0.2−43)	(10.6−13.7)	[40]
*Cronobacter sakazakii*	60	(0.9−4.4)	5.6	[41]
*E. coli* STEC	60	(0.5−3)	(3.5−7)	[42]
*L. monocytogenes*	65	(0.2−2)	(4−11)	[43]
*Salmonella* spp.	60	(2−6)	6.5	[44]
*S. aureus*	60	(0.8−10)	7	[45]
*S. aureus* (enterotoxin)	121	(8.3–34)	(25−33)	[46]

* Values considered for calculation were the worst case, meaning the higher D_ref_ and z.

**Table 4 foods-12-00572-t004:** Proximate composition and a_w_ of fresh *Tenebrio molitor* (TM) larvae and different powders (Means ± standard deviation).

	Alive Larvae	Powder A	Powder B	Powder C	Powder D
**Dry matter** (DM, g/100 g)	29.48 ^d^ ± 0.08	97.95 ^a^ ± 0.02	97.87 ^a^ ± 0.13	95.54 ^c^ ± 0.18	96.83 ^b^ ± 0.15
**Crude protein** (%, DM)	66.08 ^b^ ± 0.41	57.90 ^cd^ ± 1.01	70.04 ^a^ ± 1.42	58.37 ^c^ ± 0.36	55.62 ^d^ ± 0.56
**Crude fat** (%, DM)	19.94 ^bc^ ± 1.33	28.21 ^a^ ± 0.91	16.84 ^c^ ± 0.23	23.63 ^ab^ ± 4.19	24.51 ^ab^± 0.32
**Ash** (%, DM)	4.60 ^a^ ± 0.04	3.14 ^e^ ± 0.09	3.71 ^c^ ± 0.06	3.44 ^d^ ± 0.14	4.03 ^b^ ± 0.11
**Carbohydrates** (%, DM)	9.38 ^b^ ± 1.67	10.75 ^ab^ ± 1.56	9.41 ^b^ ± 1.61	14.55 ^ab^ ± 4.01	15.84 ^a^ ± 0.79
**a** _w_	0.99 ^a^ ± 0.00	0.08 ^c^ ± 0.00	0.16 ^b^ ± 0.00	0.17 ^b^ ± 0.01	0.17 ^b^ ± 0.00

Different letters within the same row indicate significant (*p* < 0.05) differences between means.

## Data Availability

Data is contained within the article or Appendix A.

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
