# Peer review of "Quality of Tenebrio molitor Powders: Effects of Four Processes on Microbiological Quality and Physicochemical Factors"

_foods, 2023, doi:10.3390/foods12030572_

Round 1

Reviewer 1 Report

Hi dear the editorial boards and the authors

This article "Quality of Tenebrio molitor Powders: effects of four processes on microbiological quality and physicochemical factors” was revised and has a novelty and I recommend it for after consideration of the following comments.

·        Please consider the background of mealworm privileges in abstract.

·        In Abstract the sentence of “four different processing pathways (with boiling or freezing slaughter, oven- or freeze-drying, defatting or not, grinding etc.)” why four pathways. This text can cause 16 factorial pathways. Therefore please point it as detail. The text is very vague.

·         Please consider the results related to the” dry matter, protein, fat, ash, water activity (aw)” as a detail expression in abstract.

·        The type of statistical design used in this research should be mentioned in abstract.

·         Please point to the four treatment as detail in the last paragraphs of introduction.

·         Why were these treatments chosen, for example, why other drying treatments such as fluidized bed dryer were not used and the very expensive freeze drying treatment was used?

·          Please write materials as Company Name (City, Country), especially for chemical analysis assessment which used in the study.

·        “2.2. Insect powder processing trial” paragraph is very complexity. Please express it very clear and simple.

·        The way of expressing the method of measuring macronutrients and other parameters has a scientific flaw. Please take help from the following article for the correct way of expressing it, so that the standard number of the working method should be clearly stated (https://doi.org/10.1590/fst.60820).

·        All Tables and figures e.g. Figure 2 and 3: The alphabetical statistical letters for the means should all be modified such that the greatest number has the letter a and as the numbers go lower, letters b, c etc.

·        Table 3: It seems that the statistical comparison of Alive Larvae with four types of powders, especially for dry meter and aw, increases the statistical error for four types of produced powders. It is better to compare only four types of powder separately.

·        Discussion text must grammar improve and in some cases it is very weak and maybe there is no discussion at all.

·        Conclusion is very general, try to make it more scientific, comprehensive and concise in detail, especially.

Author Response

We thank the reviewer for the relevant suggestions and we acknowledge the time and effort on this article. Your comments have helped us a lot to improve our manuscript. You will find below a point-by-point answer and changes are highlighted in yellow in the manuscript file.

Reviewer 2 Report

Title: Quality of Tenebrio molitor Powders: effects of four processes on microbiological quality and physicochemical factors

The manuscript “Quality of Tenebrio molitor Powders: effects of four processes on microbiological quality and physicochemical factors” determined the microbiological quality and physicochemical properties of Tenebrio molitor (mealworm) powder, and evaluated the impact of four different processing pathways (with boiling or freezing slaughter, oven- or freeze-drying, defatting or not, grinding etc.). This study also validated the efficacy of a boiling step as critical control points (CCPs) of insect powder processing, providing primary data for the implementation of HACCP plans in the edible insect sector. It is well written article with some interesting findings; however, there are some corrections :

It is well-established to add line numbers before submitting any manuscript to any journal, otherwise, it is not easy for the reviewers to review the paper.

Rephrase the sentence: “Microbial loads of fresh mealworms were high (mean total aerobic counts 8.4 log CFU/g), but significant hazards including Salmonella spp., L. monocytogenes and C. sakazakii were not detected”.

Add significant levels or p-value within the abstract portion.

It is also recommended to add a sentence on material and methods within the abstract to make it clear for the readers.

Split the second paragraph of the introduction part into two. I would suggest to split it from “Recent studies have shown that edible insects generally contain complex ecosystems with marked variations in microbial load and diversity as well as stable and species-specific microbiota”.

Add the name of the French company specialized in rearing insects for pet food?

Add city and country of the blender (Russell Hobbs).

Replace the reference number 3 with latest one.

Overall, in discussion part is good, however, authors should discuss more about the physicochemical properties of fresh and powdered mealworms, which is also an important aspect.

In conclusion part, authors discussed the results, I recommend to rewrite the conclusion part to conclude the study rather then again discussing the results. Secondly, authors should add one paragraph as futuristic aspects about using the Tenebrio molitor Powder as meal. Authors should also suggest to conduct studies on sensory evaluation and overall acceptance of the powder.

English grammar and sentence structure should be revised and corrected throughout the manuscript.

Author Response

(The authors gave the same response as above.)

Round 2

Reviewer 2 Report

The manuscript is sufficiently improved to be accepted for the publication.